# Built structures influence patterns of energy demand and CO$_2$ emissions across countries

Helmut Haberl [1,5] ✉, Markus Löw[1,5], Alejandro Perez-Laborda[2,5], Sarah Matej [1], Barbara Plank [1], Dominik Wiedenhofer [1], Felix Creutzig [3,4], Karl-Heinz Erb [1] & Juan Antonio Duro[2]

Built structures, i.e. the patterns of settlements and transport infrastructures, are known to influence per-capita energy demand and CO$_2$ emissions at the urban level. At the national level, the role of built structures is seldom considered due to poor data availability. Instead, other potential determinants of energy demand and CO$_2$ emissions, primarily GDP, are more frequently assessed. We present a set of national-level indicators to characterize patterns of built structures. We quantify these indicators for 113 countries and statistically analyze the results along with final energy use and territorial CO$_2$ emissions, as well as factors commonly included in national-level analyses of determinants of energy use and emissions. We find that these indicators are about equally important for predicting energy demand and CO$_2$ emissions as GDP and other conventional factors. The area of built-up land per capita is the most important predictor, second only to the effect of GDP.

Increasing global temperature driven by growing greenhouse gas (GHG) emissions is a major global concern[1]. GHGs mainly result from the energy-related combustion of fossil fuels[2]. The question emerges, which factors drive energy demand and emissions, and to what extent they are malleable. This question is relevant at many levels, from products to individuals, households, and cities up to continental and even global totals. We here focus on countries, because many decision-making processes occur at the national level. A widespread approach for cross-country analyses classifies the factors influencing resource use and emissions into population, affluence, and technology[3–5]. These factors are captured in the STIRPAT (Stochastic Impacts by Regression on Population, Affluence and Technology) framework[3], derived from the classical IPAT (impact = population × affluence × technology) approach[5,6].

Economic activity (called affluence in IPAT-style analyses, usually measured as Gross Domestic Product or GDP) is acknowledged as a major determinant of energy use and greenhouse gas (GHG) emissions[7–9]. The discussion mainly focuses on whether GDP can be decoupled from emissions, i.e., whether energy use and emissions can be reduced while GDP is growing. This may be possible e.g., through more efficient technologies, but the debate is so far inconclusive[10,11]. While recent studies revealed examples of growing national economies where policies implemented in the last decade achieved reductions in energy demand and CO$_2$ emissions[12–14], neither those studies nor a meta-analysis[15] yielded evidence for reductions in energy use and GHG emissions consistent with ambitious climate targets.

Other potential determinants of energy demand and emissions receive less attention than GDP, even though additional entry points to accelerate decarbonization are urgently required. Population density has been studied with varied outcomes. An econometric study of OECD countries 1980–2011 suggested an inverse relation between population density and CO$_2$ emissions[16]; a regression analysis of materials used in >100 countries in 2000 found a similar effect[17]. A global regression analysis[18] found no effect of population density on energy demand, whereas a panel analysis of 11 Asian countries from 1960–2004[19] to conditions that influence energy demand for heating or cooling of buildings[20], fuel prices affecting demand for transport energy[21], and the urbanization rate (urban population as percent of the

[1]Institute of Social Ecology, University of Natural Resources and Life Sciences, Vienna, Austria. [2]Economics Department and Eco-SOS, Universitat Rovira i Virgili, Tarragona, Spain. [3]Mercator Research Institute on Global Commons and Climate Change, EUREF 19, 10829 Berlin, Germany. [4]Technical University Berlin, Straße des 17 Junis 135, 10623 Berlin, Germany. [5]These authors contributed equally: Helmut Haberl, Markus Löw, Alejandro Perez-Laborda. ✉e-mail: helmut.haberl@boku.ac.at

total population). A suitable proxy for the climate dependency of heating energy demand are heating-degree days (abbreviated as HDD)[20]. The price of gasoline (abbreviated as PGAS) is an energy price indicator that is strongly related to settlement patterns[21]. Some studies used urban population as a percent of the total population (abbreviated as UPOP) as a development indicator[22].

At the urban scale, the influence of population density and the spatial layout of urban areas on cities' resource demand has been widely studied[20,23–28]. There are several reasons why the extent and spatial layout (density and form) of built structures–henceforth denoted as material stock patterns[29], i.e., the spatial patterns of societies' material stocks in infrastructures and buildings–could affect energy demand and $CO_2$ emissions. The accumulation of material stocks requires massive amounts of resources such as steel-reinforced concrete, mortar, bricks, timber, plastics, glass, gravel, or sand[30–33], which are associated with high GHG emissions[34]. Heating, cooling, and lighting of buildings and production processes in industrial plants require much energy[20,23,26,35], as does the mobility of goods and people on roads and railways[33,36].

Despite these insights from urban studies, material stock patterns are seldom considered in debates on national-level analyses of factors determining levels of energy demand and emissions, as well as their possible decoupling from GDP[11,15]. A systematic review of the empirical literature on these questions[15,37] revealed only one study[38] considering material stock patterns in analyzing transport-related emissions. Hence little is known about the effects of material stock patterns on energy demand and $CO_2$ emissions beyond the city level. This results from a scale mismatch: maps of material stock patterns provide fine-grained spatial detail[30], often focused on specific regions, that cannot be included in national-level analyses of factors driving energy demand and emissions. For cross-country analyses, consistent indicators need to be developed from spatial data and then aggregated at the national level in a manner which preserves key information on patterns and supports comparative analyses across countries and world regions.

In this work, we develop national-level indicators of characteristics of built structures that, based on urban studies, can be potential determinants of resource use and emissions. We quantify them for 113 countries comprising 91.2% of the world population and 97.3% of global GDP. We analyze two independent variables: (1) yearly total per-capita final energy consumption (abbreviated as TFC) and (2) yearly per-capita $CO_2$ emissions (abbreviated as $CO_2$). We test material stock pattern indicators against other variables that have been widely used in national-level analyses of determinants of energy use and emissions, here denoted as conventional factors. As conventional factors, we use GDP/cap/yr (abbreviated as GDP), population density (DENS), UPOP as a development indicator, HDD as a proxy of climate dependency of energy demand, and PGAS as an energy price indicator. Extensive variables are expressed as per-capita values to facilitate country comparisons and remove countries' population numbers from the analysis.

## Results

### National-level indicators of material stock patterns

We test three hypotheses based on aggregated indicators of material stock patterns (Fig. 1). Material stock patterns are represented by three types of indicators: (1) The area of built-up land is represented by two indicators, one as a fraction of a nation's inhabited land, the other per inhabitant. Other indicators describe patterns of built-up areas, including their spatial clustering, form, and distribution, which reflect geomorphological factors as well as historical contingencies. (2) Road indicators that describe the density (length per unit area) of roads in urban and rural regions and the relations between urban and rural road lengths and densities. (3) Railway indicators are defined in the same manner as those for roads (Table 1).

### Bivariate and semi-partial correlations

In terms of their Pearson coefficients in bivariate correlations, several indicators of material stock patterns are as strongly correlated with TFC and $CO_2$ as the conventional factors (Fig. 2a). GDP is positively correlated with both TFC and $CO_2$. HDD and the fraction of the urban population also show the expected pattern, while PGAS and DENS are largely uncorrelated. Almost all material stock pattern indicators are correlated with both TFC and $CO_2$. The extent of built-up land ($BL_{cap}$ and $BL_{fract}$) is positively correlated with both TFC and $CO_2$, as are total and rural road density and the dispersion of built-up land and most railway-related indicators. The correlation coefficients of $BL_{cap}$ with TFC and $CO_2$ are both ~0.7; $BL_{cap}$ is the second-best predictor of both TFC and $CO_2$ after GDP. As expected from the urban literature, urban population density ($UP_{dens}$) is inversely correlated with $CO_2$ and energy, whereas the share of urban population (UPOP) is strongly positively correlated with $CO_2$ and TFC. Inverse relations prevail for $BL_{mono}$, $BL_{comp}$, and the urban-to-rural relations of infrastructure density.

Semi-partial correlations of the material stock pattern indicators controlled for GDP and population density (DENS) are shown in Fig. 2b. The part of each material stock pattern indicator correlated with GDP and DENS is removed, revealing the strength of the linear correlation between TFC or $CO_2$ and the remaining part of the respective variable. The distance from the vertical axis either to the right (positive correlation) or to the left (inverse correlation) depicts the additional explanatory power of the respective indicator over a model considering only GDP and DENS. Several indicators provide additional explanations over GDP and DENS alone. The area of built-up land (both $BL_{cap}$ and $BL_{fract}$) is positively correlated with TFC and $CO_2$, as are the density of rail and road infrastructures, especially in rural areas. UPOP is inversely correlated with TFC and $CO_2$, as is urban road density (not significant) and $BL_{mono}$. PGAS, which had not been significant in the bivariate correlations, emerges as an important factor, which is also observed for $RD_{urban}$ and other indicators. $UP_{dens}$ lose importance, most likely due to its high correlation with GDP.

### Multivariate lasso analysis

The capability of the material stock pattern indicators to add insights beyond conventional factors is further analyzed in Table 2. We use the least absolute shrinkage and selection operator (lasso) approach to select variables for multivariate statistical models capable of predicting cross-country patterns of TFC and $CO_2$. Lasso is a widely-used procedure for automatically performing variable selection in linear regression models[39,40]. It overcomes the drawbacks of overfitting and multicollinearity associated with ordinary least square (OLS) methods[41]. The lasso method penalizes complexity to derive the best parsimonious model for any predefined value of λ (the factor determining how strongly model complexity is penalized; for detail, see Methods section), thereby allowing to identify the factors that most strongly affect TFC and $CO_2$. The standard procedure to select λ is cross-validation, where randomly chosen samples of countries are used to develop a model used for the prediction of patterns in out-of-sample countries. The procedure gradually reduces λ, which generally results in more indicators being selected[41]. Alternative Lasso schemes and other variable selection techniques, such as stepwise regression, yield very similar results (SI).

The leftmost three columns of Table 2 show the knots in the lasso paths, i.e., the λ values at which indicators are added (or removed due to collinearity). Knots are arranged in decreasing order of λ, with indicators being ranked in order of selection. In the first column, the λ value marked with an asterisk (*) indicates the optimal model identified by the cross-validation. GDP is always selected as the first indicator, but various material stock pattern indicators appear very early on (i.e., at high λ values), and remain active in the optimal model. $BL_{cap}$ is the second-chosen indicator for both $CO_2$ and TFC, and several other

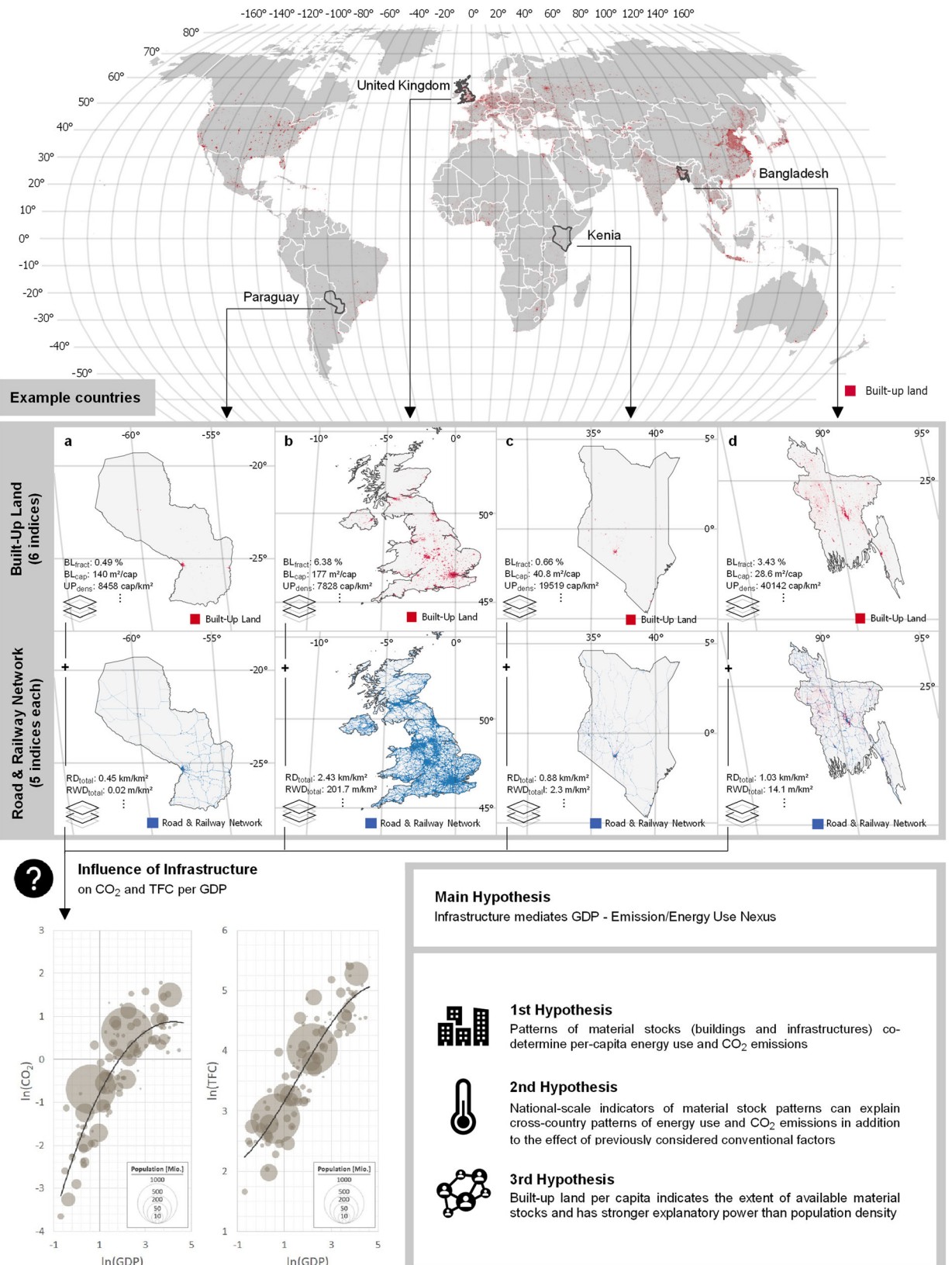

**Fig. 1 | Workflow of this study.** National-level indicators of the extent and spatial patterns of settlements and infrastructures (material stock patterns) were derived from global maps, here illustrated using Paraguay, the UK, Kenia, and Bangladesh as examples. Results were statistically analyzed along with the conventional factors assumed to co-determine energy use and $CO_2$ emissions. The main aim was to test the hypotheses formulated in the lower-right box. Copyright for administrative boundaries: © Eurogeographics.

**Table 1 | Indicators of the extent and patterns of built-up land and transport infrastructures**

| Name | Abbreviation | Description and interpretation | Unit |
|---|---|---|---|
| **(1) Indicators for the extent and pattern of a nation's built-up land** | | | |
| Fraction of built-up land | $BL_{fract}$ | Built-up land (buildings & infrastructures) as % of the inhabited land area. | $m^2/m^2$ |
| Built-up land per capita | $BL_{cap}$ | Built-up land per capita. | $m^2/cap$ |
| Dispersion of built-up land | $BL_{disp}$ | Index based on the average distance of each patch of built-up land to the nearest adjacent patch. High values indicate strong dispersion. | – |
| Monocentricity of built-up land | $BL_{mono}$ | Area of the largest contiguous built-up patch as % of the sum of the areas of the ten largest patches. High values indicate dominance of one large center. | $m^2/m^2$ |
| Compactness of built-up land | $BL_{comp}$ | Index describing how round or compact the shapes of a nation's built-up land patches are on average. | – |
| Urban population density | $UP_{dens}$ | Urban population per unit of urban built-up land | $cap/m^2$ |
| **(2) Indicators of road density and distribution** | | | |
| Road density | $RD_{total}$ | Length of roads per unit area of inhabited land. | $m/m^2$ |
| Urban road density | $RD_{urban}$ | Length of roads in urban areas per unit area of urban areas. | $m/m^2$ |
| Rural road density | $RD_{rural}$ | Length of roads in rural areas per unit area of rural areas; proxy of rural accessibility and connectivity between urban centers. | $m/m^2$ |
| Ratio of urban-to-rural road lengths | $RL_{urb-rur}$ | Ratio of urban to rural road lengths, indicating the extent to which roads are concentrated in cities. | – |
| Ratio of urban-to-rural road density | $RD_{urb-rur}$ | Ratio of $RD_{urban}$ and $RD_{rural}$, indicating the difference between urban and rural road densities. | – |
| **(3) Indicators of railway density and distribution** | | | |
| Railway density | $RWD_{total}$ | Length of railways per unit area of inhabited land. | $m/m^2$ |
| Urban railway density | $RWD_{urban}$ | Length of railways in urban areas per unit area of urban areas. | $m/m^2$ |
| Rural railway density | $RWD_{rural}$ | As $RD_{rural}$ for railways. | $m/m^2$ |
| Ratio of urban-to-rural railway lengths | $RWL_{urb-rur}$ | As $RL_{urb-rur}$ for railways. | – |
| Ratio of urban-to-rural railway density | $RWD_{urb-rur}$ | As $RD_{urb-rur}$ for railways. | – |

These indicators condense spatially explicit information in maps to national-level indicator values assumed to co-determine a nation's per-capita level of energy demand and $CO_2$ emissions. Global maps showing the indicator values are in Supplementary Figs. 5–7 (SI).

material stock pattern indicators are selected much earlier than widely-used conventional factors such as population density (DENS). Heating-degree days (HDD) and the price of gas (PGAS) are important for both $CO_2$ and TFC, whereas UPOP is important for predicting $CO_2$ but not selected by lasso for TFC.

The rightmost three columns of Table 2 report the estimated coefficients of models selected by cross-validation as well as measures of in-sample and out-of-sample fit ($r^2$). Models A comprises only variables selected by lasso (path shown in the left part of the table) among all indicators. For the benchmark Models B, only conventional factors are selected. Including material stock pattern indicators in Model A yields better prediction than Models B for both $CO_2$ and TFC in terms of in-sample and out-of-sample goodness of fit, and improves the Bayesian Information Criterion (BIC) of model selection. Other criteria to develop optimal models, and alternative variable selection procedures yielded similar results (section 5 in the SI). Even models that include only material stock pattern indicators and exclude all conventional factors (including GDP) achieve good predictions (out-of-sample $r^2$ of 0.65 for TFC and 0.62 for $CO_2$; see results for Model C shown in section 5.2 of the SI).

## Discussion

The analysis shows that extent and spatial patterns of built structures, here denoted as material stock patterns, play an important role in co-determining and predicting the level of resource use, here TFC and $CO_2$, in a cross-country analysis. This implies that insights from urban studies[23–26,42] generally hold at the national scale. Despite the unavoidable loss of information resulting from the aggregation of maps to the national scale, the indicators in Table 1 maintain key information representing important characteristics of material stock patterns that strongly affect cross-country patterns of TFC and $CO_2$. The indicators have substantial additional explanatory and predictive power over conventional factors. They can help develop much stronger models of national-level TFC and $CO_2$ than usual IPAT-

type approaches, and will enable researchers to broaden their analysis and scenario modeling capabilities by including material stock patterns as crucial factors for analyzing the possible decoupling of energy use and emissions from GDP.

Population density plays a smaller role than widely assumed, while many material stock patterns strongly influence the cross-country differences in energy demand and $CO_2$ emissions. The material stock pattern indicator with the most consistent predictive power across all analyses is the area of built-up land per capita ($BL_{cap}$), which emerges as the second-most important variable (after GDP) in most of our statistical analyses, even in analyses considering the GDP effect. This is plausible because infrastructures and buildings require energy for being built and used, which results in $CO_2$ emissions in current fossil fuel-dominated energy systems[34,43]. Higher $BL_{cap}$ also means larger floor size and longer distances between destinations, which all raises energy demand in buildings and transport. These findings corroborate and expand on previous analyses (that used entirely different models, did not capture spatial patterns, and mostly referred to temporal trends), suggesting that challenges for climate-change mitigation strongly depend on the past and future accumulation of material stocks in buildings and infrastructures[43–46]. This is worrying because material stocks are growing globally largely in unison with GDP[32,46].

Our indicators and results create options for analyzing which of the characteristics of material stock patterns are most important in determining and predicting energy demand and emissions in cross-country analyses. Analyses presented in Fig. 2, Table 2 as well as the SI clearly show that many specific aspects of material stock patterns play a role in determining cross-country differences in energy demand and emissions. The multivariate analysis also shows that these patterns interact in many ways that are difficult to disentangle due to the collinearities of the material stock pattern indicators. Future research could employ refined study designs addressing how changes over time and space affect these relationships and elucidate the different causal pathways involved.

(a)

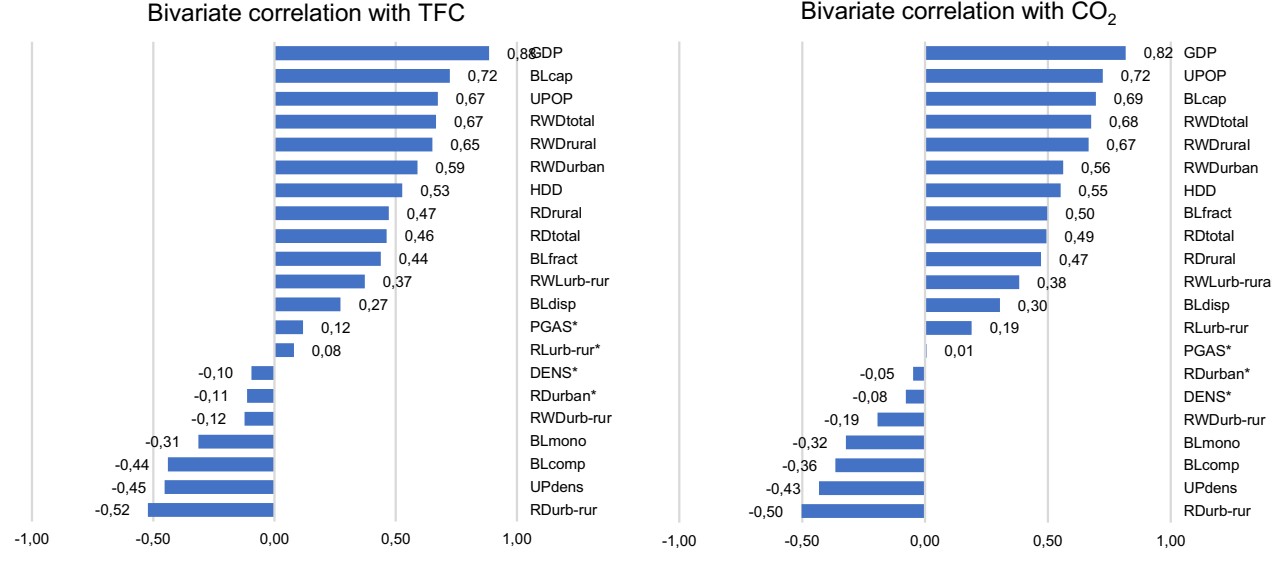

(b)

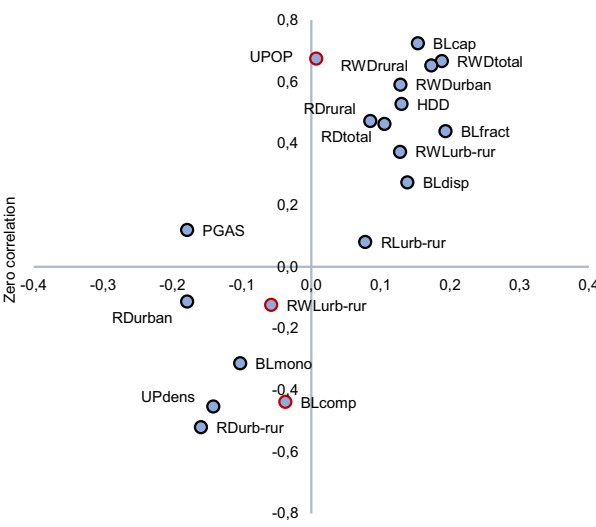
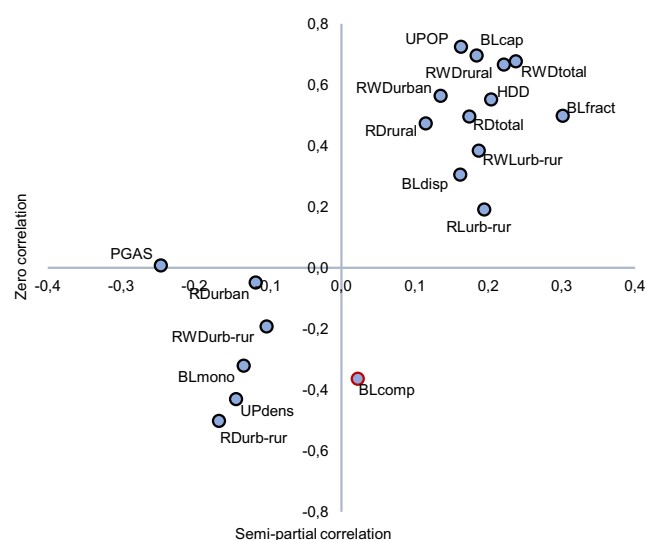

**Fig. 2 | Correlation analyses of total final energy demand per capita (TFC) and per-capita CO₂ emissions (CO₂) with conventional factors and material stock pattern indicators. a** Pearson's zero correlation coefficients of correlations between TFC (left) and CO₂ (right) and material stock pattern indicators as well as conventional factors. Natural logarithms of the variables were analyzed. Squaring the correlation coefficients gives the percentage of the cross-country variation of CO₂ or TFC explained by the respective indicator alone. Correlations were not significant for variables marked with an asterisk ($p < 0.1$). **b** Semi-partial correlations between material stock pattern indicators and TFC (left) and CO₂ (right) controlling for GDP and DENS. Distance from the vertical axis indicates the correlation coefficient of the semi-partial correlation, and distance from the horizontal axis is the correlation coefficient of the bivariate (uncontrolled) correlation. Red contours indicate insignificant results (significance level $p < 0.1$).

Our results have implications for countries pursuing ambitious climate targets[2,43]. They demonstrate that insights from urban-level studies translate to the national scale, which could so far not be investigated at that level due to lacking data. This suggests that the indicators presented above offer opportunities to investigate the importance of built structures also for other aspects of national economies' resource use than those analyzed here. Empirical urban studies have consistently shown that urban form and infrastructure affect travel demand and, therefore, GHG emissions[47]. The relationship between built structures and GHG emissions has also been predicted by theoretical analyses in urban economics[21,48]. This suggests that our national-level observations—e.g., that large, sprawling material stocks in built structures represent an important determinant of a nation's per-capita level of energy demand and emissions—are very likely underpinned by similar causal relationships. Of course, more research, in particular on how these relationships change over time, would be desirable. Because high-resolution maps of decadal trends in built structures are gradually becoming available[49,50], such analyses may

**Table 2 | Multivariate analysis using cross-validation lasso**

| **Models predicting TFC** | | | | | |
| --- | --- | --- | --- | --- | --- |
| **Lasso path for model A** | | | **Estimated coefficients** | | |
| $\lambda$ | **(A)dded, (R)emoved** | **CV MPSE** | | **Model A** | **Model B** |
| 0.711 | GDP(A) | 0.703 | GDP | 0.494 | 0.572 |
| 0.371 | $BL_{cap}$(A) | 0.319 | DENS | −0.017 | −0.039 |
| 0.161 | HDD(A), $RWD_{tot}$(A) | 0.179 | UPOP | | |
| 0.121 | $RD_{urb-rur}$(A) | 0.166 | HDD | 0.021 | 0.051 |
| 0.111 | PGAS(A) | 0.161 | PGAS | −0.352 | −0.288 |
| 0.101 | $RD_{urb}$(A) | 0.155 | $BL_{cap}$ | 0.184 | |
| 0.076 | $RWD_{urb}$(A) | 0.135 | $BL_{mono}$ | −0.017 | |
| 0.063 | $RWD_{tot}$(R) | 0.125 | $RD_{urb}$ | −0.392 | |
| 0.048 | DENS(A) | 0.115 | $RL_{urb-rur}$ | −0.024 | |
| 0.030 | $BL_{mono}$(R) | 0.108 | $RD_{urb-rur}$ | −0.074 | |
| 0.027 | $RL_{urb-rur}$(A) | 0.107 | $RWD_{urb}$ | 0.045 | |
| 0.017* | (Unchanged) | 0.105 | Intercept | 3.457 | 2.542 |
| 0.016 | $RWD_{urb-rur}$(A) | 0.105 | | | |
| 0.013 | $BL_{comp}$(A) | 0.105 | **Measures of in-and-out-of-sample fit** | | |
| 0.011 | (Unchanged) | 0.106 | BIC | 84.17 | 100.32 |
| | | | $r^2$ | 0.900 | 0.851 |
| | | | $oSr^2$ | 0.865 | 0.833 |

| **Models predicting CO₂** | | | | | |
| --- | --- | --- | --- | --- | --- |
| $\lambda$ | **(A)dded, (R)emoved** | **CV MPSE** | **variable** | **Model A** | **Model B** |
| 0.867 | GDP(A) | 1.278 | GDP | 0.449 | 0.582 |
| 0.544 | $BL_{cap}$(A) | 0.801 | DENS | −0.061 | 0.023 |
| 0.496 | UPOP(A) | 0.738 | UPOP | 0.466 | 0.558 |
| 0.452 | $RWD_{tot}$(A) | 0.683 | HDD | 0.055 | 0.109 |
| 0.312 | HDD(A) | 0.530 | PGAS | −0.680 | −0.688 |
| 0.215 | PGAS(A) | 0.446 | $BL_{fract}$ | 0.201 | |
| 0.135 | $RD_{urb-rur}$(A) | 0.338 | $BL_{cap}$ | 0.176 | |
| 0.123 | $BL_{fract}$(A) | 0.323 | $BL_{comp}$ | 0.489 | |
| 0.085 | $RWL_{urb-rur}$(A) | 0.289 | $RD_{urb}$ | −0.201 | |
| 0.070 | $BL_{fract}$(R) | 0.280 | $RWD_{urb}$ | 0.130 | |
| 0.048 | $BL_{comp}$(A) | 0.269 | $RWD_{rur}$ | 0.017 | |
| 0.044 | $RD_{urb}$(A) | 0.266 | Intercept | −2.723 | −4.294 |
| 0.037 | $RWD_{urb}$(A) | 0.262 | | | |
| 0.025 | $BL_{fract}$ (A) | 0.256 | **Measures of in-and-out-of-sample fit** | | |
| 0.023 | DENS(A) | 0.255 | BIC | 178.44 | 190.82 |
| 0.021 | $RWL_{urb-rur}$(R) | 0.254 | $r^2$ | 0.873 | 0.812 |
| 0.016 | $RWD_{rur}$(A) | 0.251 | $oSr^2$ | 0.817 | 0.785 |
| 0.016 | $RD_{urb-rur}$(R) | 0.251 | | | |
| 0.014 | $RWD_{tot}$(R) | 0.250 | | | |
| 0.012* | (Unchanged) | 0.249 | | | |
| 0.006 | (Unchanged) | 0.250 | | | |

The leftmost three columns show the lasso path for predicting cross-country patterns of TFC (above) and $CO_2$ (below) using all variables. In the first column, the $\lambda$ value marked with an asterisk denotes the optimal model (Model A) emerging from the cross-validation. CV MPSE is the cross-validated mean-square prediction error evaluated with tenfolds (for detail, see Methods section). For Model B, only conventional factors are selected. The same folds are used for the assessment of all models. BIC is the Bayesian Information Criterion for model selection. $r^2$ is the goodness of fit within the sample of countries, and $oSr^2$ refers to the (cross-validated) out-of-sample goodness of fit.

soon be feasible. We conclude that the area and patterns of built-up land emerge as an important entry point for efforts at reducing energy demand and $CO_2$ emissions at national levels[51], suggesting that limiting built-up area per capita could be a worthwhile policy goal, not only from a land-use perspective, but also to limit future GHG emissions.

## Methods

The material stock pattern indicators were derived from crowd-sourced data on infrastructures (roads and railways)[52] as well as high-resolution land-cover data[53]. The indicators refer to the years 2015 (built-up land) and 2020 (roads and railroads). To reduce random fluctuations for the energy and emission data as well as the conventional factors, we calculated averages for as many years in the period 2015-2020 as were available in the statistical sources. Extensive variables were represented by per-capita values to facilitate comparisons between different-sized countries, following the convention to regard population as a scaling factor with an elasticity of one in STIRPAT analyses[5].

## Data used to derive spatial indicators

The spatial indicators presented in this study rely on three spatially explicit datasets; (1) built-up features and urban agglomerations, (2) main infrastructure features (road and railway), and (3) an inhabited land area layer used for reference. All variables that express quantities as fractions of a country's area, relate to inhabited land. We used the LC100 grid of the Copernicus Global Land Cover Service[53] to derive relevant land cover information. The choice of input data and data quality checks are discussed in the SI; see also Supplementary Fig. 1 (SI). In contrast to other built-up land datasets[54–56], the LC100 not only provides globally consistent information on built-up areas, it also includes all other complementary land cover types at a global scale. For data preparation, we vectorized the LC100 grid to allow further spatial intersection procedures (e.g., to clip to national borders). The NOAA-DEM grid was used as a global digital elevation model. We utilized the Geofabrik-download hub[57] to obtain the entire global OSM data. The GISCO archive from EUROSTAT[58] provides the applied country borders. Details on the data sources are given in Supplementary Table 1 (SI); datasets[59] and software code[60] are available online.

## Sources of energy and emission data as well as conventional factors

Gross domestic product (GDP) in constant 2015 US$ was sourced from UN Statistics Division National Accounts[61]. Population data used to calculate population density (DENS) and urban population rates (UPOP) were taken from official census data 2017 of the World Bank[62,63]. The World Bank database was also used to source data on pump prices for gasoline (PGAS)[64]. Heating-degree days (HDD) were calculated as a population-weighted average of °C days above 18 °C and sourced from the International Energy Agency (IEA) Weather for Energy Tracker[65]. Territorial $CO_2$ emissions from fossil fuel combustion and cement production ($CO_2$) were sourced from the Global Carbon Project database[66,67] and total final energy consumption (TFC) from the IEA energy balances, which could only be accessed for the latest year 2017[68]. The record date for the download of these indicators was 25 March 2021. We always used the latest available year for our cross-sectional analysis, but also conducted robustness checks using arithmetic averages over the latest five years for all indicators to exclude potential bias from outstanding annual values. Details on the data are available in Supplementary Table 2 in the SI.

## Deriving the built-up land layer

The built-up land vector (BL) for every country is one primary data product derived by the vectorized LC100 grid. This standard BL layer comprised all national built-up features and was used to derive the BL-related indicators shown in part (1) of Table 1. To map each country's urban agglomerations, and thereby distinguish urban from rural land respectively infrastructures (see Supplementary Figs. 2–4, SI), we clustered features of the BL vector layer using an empirical growing-neighborhood approach: We started with the country's largest BL feature and created convex hulls, which were buffered with the fifth of the area-equal radius. We then identified intersecting BL features within this buffered hull and created a new hull and buffer. These two steps were repeated as long as the BL area of all intersecting features reached 33.3% of the area of the buffered hull (used to check the intersections), and the BL area increased at least 0.5%, compared to the BL area of the previous iteration. If these criteria were not given, the growing procedure terminated and the collected BL features dissolved to one BL agglomeration. The algorithm subsequently went on with the next-biggest BL feature, which had not been assigned to an already-created agglomeration, again starting the growing-neighborhood procedure. The whole process terminated when the next BL feature (to restart the growing-neighborhood procedure again) represented less than 0.1% of the total BL area of the respective country. The maps

in Supplementary Fig. 5 (SI) show the built-up (BL) land indicators. Fractions of land areas refer to inhabited land.

## The infrastructure layers

The preprocessed OSM database comprises globally consistent road (R) and railway vector data (RW), but the availability of OSM data varies strongly between countries. While the OSM data in countries of the global North (industrial or even postindustrial countries) also include minor road and train track categories (e.g., cycleways, steps, or private gauges), OSM data in countries of the global South only comprise the main road and railway network. To reduce this inconsistency, we excluded minor OSM classes in order to derive a more comparable global database (Supplementary Table 3). The maps showing the road network indicators are in Supplementary Fig. 6 (SI). Fractions of the area refer to an inhabited land. The railway indicators were derived from OSM in the same manner as those for roads and are shown in Supplementary Fig. 7 (SI). The railway types considered are defined in Supplementary Table 4. Fractions refer to an inhabited land. The planar extent of road and railway networks was calculated using width data reported in Supplementary Table 5. The distinction between urban and rural infrastructures was based on a spatial intersection of OSM road and railway data with the BL features, which resulted in layers of urban and rural networks (example shown in Supplementary Fig. 4, SI).

## Reference layer for inhabited land

The definition of some material stock pattern indicators requires a reference area ($A_{REF}$). The area of the total national territory ($A_{NT}$) may not be suitable, given that in some countries (almost), the entire area is inhabitable, whereas other countries contain large tracts of land unsuitable for human habitation and hence largely uninhabited. We, therefore, developed the proxy layer inhabited land (IH) as a more suitable area reference ($A_{IH}$). In contrast to existing similar datasets[69], this IH mainly uses land cover information of the LC100 grid. This guarantees thematic consistency in spatial intersections with BL data that were derived using the same dataset. The IH includes not only area that is covered with settlements or infrastructures, but also cropland areas and areas with ambiguous land cover that fall within a zone of influence around existing built-up areas, which is approximated by a buffer that depends on the area of a built-up land feature (Eq. 1). The IH is based on the current settlement and cropland extent, as well as the spatial distribution, density, and elevation occurrence of built-up areas according to the LC100 grid.

To calculate IH, we first cut out high-altitude regions from the country's total territory. We calculated the elevational distribution of BL features and excluded all areas above the area-weighted 99th percentile of BL-elevations. In a second step, we excluded the LC100 land cover types bare/sparse vegetation (deserts and rocks), moss & lichen, snow & ice, and permanent water bodies. Thirdly, we spatially intersected the map resulting from the previous two steps with a synthetic layer that represents the gapless inhabited land area. To derive this layer, we applied an area-dependent buffer for all BL features. See Eq. (1) for the dynamic BL buffer width ($w_{BL}$); $A_{BL\ feature}$ denotes the area of an exemplary built-up land feature.

$$w_{BL} = A_{BL\ feature}^{\frac{2}{3}} \qquad (1)$$

$$\max\{w_{BL} : 100 km\}$$

Finally, we added cropland areas (from LC100) and the original BL areas to re-include those BL areas excluded by the elevation threshold. Please note that for spatial indicators that depend on $A_{REF}$, the specific spatial pattern (shape) of IH is not relevant: we just use the national total area of IH instead of the area of the national territory as reference value. The potential usefulness of this IH layer for other research

questions, in particular where its spatial accuracy is of high importance, needs to be tested and is not in focus in this study.

## Definitions of spatial pattern indicators
The spatial pattern indicators were derived from the built-up, road, and railway layers. Details on the definitions of the indicators (Table 1) are given in the SI in section 3, Supplementary Tables 6–8.

## Pearson correlations
The Pearson correlation is a measure of linear association between two variables. The coefficient can be obtained from bivariate data $\{(X_1,Y_1),...,(X_n,Y_n)\}$ as $r_{XY} = S_{XY}/S_X S_Y$, where $S_{XY}$ and $S_i$ denote the sample covariance and standard deviation. The correlation coefficient is between −1 and 1. Correlations equal to 1 (or −1) indicate a perfect linear association, with data points lying exactly on a positive (negative) line. A value equal to zero indicates the absence of any linear association. The squared correlation coefficient $r_{xy}^2$ is the coefficient of determination (R-squared) of the linear regression of variable x on variable y; it measures the fraction of the variance explained by the regression line.

## Semi-partial correlations
Suppose that $Y$ is determined by $\mathbf{X} = \{X_1,....,X_k\}$. Then, the semi-partial correlation between $Y$ and $X_i$, controlled for the other predictors $\mathbf{X}_{-i}$, attempts to measure the correlation between $Y$ and $X_i$ that would be observed if the effect of $\mathbf{X}_{-i}$ would be removed from $X_i$ but not from $Y$. This means that the semi-partial regression measures the correlation between $Y$ and the part of $X_i$ that is orthogonal to the other variables $\mathbf{X}_{-i}$. It is calculated by constructing a new variable $X_i'$ that is orthogonal (i.e., entirely uncorrelated) to all previously considered variables (i.e., those controlled for).

The squared semi-partial correlation coefficient measures the fraction of the variance of the dependent variable $Y$ that is uniquely explained by $X_i$. Thus, it can also be interpreted as the increase (decrease) in the model R-squared value that results from including (removing) $X_i$ from the set of predictors $\mathbf{X}$.

The semi-partial correlation is calculated by first fitting a linear regression of $Y$ on $\mathbf{X}$ and computing the coefficient as:

$$r_{Y(X_i|\mathbf{X}_{-i})} = \text{sign}(t)\sqrt{\frac{t^2(1-R^2)}{n-k}} \qquad (2)$$

In (2) $t$ is the t-statistic of variable $X_i$ in the previous regression, $R^2$ is the R-squared, $k$ is the number of independent variables plus the constant, and $n$ is the sample size. Finally, the significance level is given by $2/\Pr(t_{n-k} > |t|)$, where $\Pr$ is the probability, $t$ is as described above and $t_{n-k}$ follows a Student's $t$ distribution with $n-k$ degrees of freedom. Further details on the correlation techniques used are available e.g. here[70].

## Lasso analysis
The least absolute shrinkage and selection operator (lasso) approach is used to select variables for multivariate statistical models for predicting cross-country patterns of TFC and $CO_2$[41,71]. Lasso is a standard regularization technique for model selection and prediction that overcomes the disadvantages of other regularization techniques such as Ridge regression[72]. It can select a parsimonious set of variables from many potential covariates, even if covariates are collinear. Lasso minimizes the sum of squared residuals, but unlike standard least square fit approaches, it penalizes complexity in the objective function to derive the best parsimonious model for any predefined value of $\lambda$, i.e., the factor penalizing model complexity. If $\lambda$ is set to zero, lasso delivers standard least squares estimates, which corresponds to a

model with the maximum complexity. In general, the larger the $\lambda$, the smaller the number of non-zero coefficients.

Consider a linear specification $Y = B_0 + B_1 X_1 + \ldots + B_P X_P + \epsilon$, where variables have been previously standardized to account for differences in scales. Lasso finds estimates for model coefficient $\mathbf{B}$ keeping the model sparse by minimizing the following term:

$$\frac{1}{2N}(Y - \mathbf{XB})'(Y - \mathbf{XB}) + \lambda\sum\nolimits_{j=1}^{P}|B_j| \qquad (3)$$

The first part of the term (3) is the in-sample squared error minimized by a classical least-squares approach. Lasso also includes the absolute sum of coefficients in the objective function, which penalizes complexity driving some of the estimated coefficients to zero.

The value of $\lambda$ is typically chosen so that the estimated model satisfies a predetermined condition. Several criteria can be employed, the most common of which is cross-validation. Cross-validation selects the $\lambda$ to minimize the out-of-sample prediction error. First, sample observations are split into K random folds (validation sets). For each validation set, the model is fitted using data from the other folds, and the out-of-sample deviance for the observations in the validation set is computed (i.e., using data not employed for estimation). Finally, the overall out-of-sample performance of the model in all the validation sets is assessed by the mean-square prediction error (MSPE), a statistical parameter in squared (log) units required for model selection. Cross-validation selects the $\lambda$ over a grid of possible values such that the corresponding model has the minimum MSPE[71]. In Table 2, MSPE is transformed into $r^2$, i.e., the goodness of fit within the sample of countries, and $oSr^2$, i.e., the cross-validated out-of-sample goodness of fit for the optimal models. If no variable is added or removed at $\lambda^*$, this is reported in the left columns of Table 2 as Unchanged. Note that beyond $\lambda^*$, more variables could be added but would not further improve the out-of-sample prediction.

## Data availability
Datasets on spatial data on patterns of global infrastructure and settlements, the inhabited land layer, as well as the indicator values of dependent and independent variables used in the statistical analyses is freely available here: https://doi.org/10.5281/zenodo.5876941. An interim result that was too large to be uploaded as zenodo archive is available from the authors for non-commercial research purposes upon reasonable request (for detail, see ref. 59).

## Code availability
The code used for calculations of maps is freely available here: https://doi.org/10.5281/zenodo.5883652.

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

## Acknowledgements

This research has received funding from the European Research Council (ERC) under the European Union's Horizon 2020 research and innovation program, MAT_STOCKS, grant agreement No 741950, awarded to H.H., and from Grant PID2020-119152GB-I00, funded by MCIN/AEI/ 10.13039/501100011033, awarded to J.A.D.

## Author contributions

H.H. conception of research design, data analysis, interpretation of results, article drafting, project management, and funding acquisition. M.L. development of spatial indicators, spatial analyses, data analysis, quantification of indicators, designing figures, and contribution to writing. A.P.-L. conceptualization and implementation of statistical analyses, data analysis, designing figures, and contribution to article writing, S.M. spatial and statistical analyses, data analysis and interpretation, and contribution to writing, B.P. collection of STIRPAT indicators, data analysis and interpretation, and contribution to writing, D.W., F.C., and K.-H.E. contribution to research design, data analysis and interpretation, and contribution to writing, J.A.D. conception and supervision of statistical analyses, research design, data analysis and interpretation, and contribution to writing.

## Competing interests

The authors declare no competing interests
