## [Peer Review File · Nature Communications]

Built structures influence patterns of energy demand and CO2 emissions across countriesEditorial Note: This manuscript has been previously reviewed at another journal that is not operating a transparent peer review scheme. This document only contains reviewer comments and rebuttal letters for versions considered at *Nature Communications* .

REVIEWER COMMENTS

Reviewer #4 (Remarks to the Author):

I am sorry that I am still not convinced by the reviewers' response in addressing my comments.

Reviewer #5 (Remarks to the Author):

To start, let me state that as a "new" reviewer, involved in an already active discussion on the paper "Built structures strongly influence cross-country patterns of energy demand and CO₂ emissions" I really appreciate the effort the authors have done to improve the paper and replying the reviewers on several points, avoiding me the effort of reflecting on aspects already clarified or asking questions already answered.

Having said that, I think the paper now deserves publication. In my opinion the paper provides has novelty and it is the result of a serious analytical work. In particular, I join Rev #4 in appreciating the huge effort put in defining the indicator toolbox and assessing its quality.

About possible alternative datasets or other possible data analysis patterns: surely they are possible and reviewers have given relevant suggestions on this. Nevertheless, authors have made a set or rational methodological choices both in selecting data and applying statistical tools and have been very transparent in explaining the reasons of their choices. Provided that these choices are convincing and scientifically sound (as I think they are), I believe they should be respected. Of course other pathways, including the application of methods borrowed from epidemiology could have been possible, but this could be the subject of a paper from another group or even from the authors themselves in future. Here the question is if the methodology and the input data chosen answer the research question posed, and in my opinion the authors mostly succeeded in this.

Even here, my main concern has been already voiced by Rev #4 noticing: "It seems that the authors are observing the relationships between CO₂ and material stock indicators observed from the difference among countries, and then expecting that, for each country, change in the material stock pattern indicators will cause CO₂ emissions to go down. I think that the authors cannot reach this conclusion." Again, I agree with the rephrasing of this part of the paper made by the authors, that I consider acceptable for the publication.

On the contrary, I was disappointed by the very last sentence in the conclusions: "The area and patterns of built-up land emerge as important and novel entry point for efforts at reducing energy demand and CO₂ emissions, which could contribute to limiting the reliance on contested CO₂ reduction options such as bioenergy with carbon capture and sequestration" Citing BECCS in the last sentence, without providing context and summarizing a complex debate in two lines and a single citation does not do justice to the authors' capacity of analyzing complex issues they have shown throughout the paper. Indeed, it is true that BECCS can be a questionable source of energy, but it is also true that several other energy sources (I would say, EVERY energy source) has pros and cons, and I do not understand why open a view on such a complex debate in the last two lines of a good paper focused on totally

different topic. On summary, I strongly suggest to remove this sentence or to give it a more general meaning.

REVIEWER COMMENTS

Reviewer #4 (Remarks to the Author):

I am sorry that I am still not convinced by the reviewers' response in addressing my comments.

Many thanks for re-reading our manuscript and rebuttal. We regret this situation, but given the lack of a new substantive argument (or reply to our rebuttal) we were not able to address the concerns of this reviewer.

Reviewer #5 (Remarks to the Author):

To start, let me state that as a “new” reviewer, involved in an already active discussion on the paper “Built structures strongly influence cross-country patterns of energy demand and CO2 emissions” I really appreciate the effort the authors have done to improve the paper and replying the reviewers on several points, avoiding me the effort of reflecting on aspects already clarified or asking questions already answered.

Having said that, I think the paper now deserves publication. In my opinion the paper provides has novelty and it is the result of a serious analytical work. In particular, I join Rev #4 in appreciating the huge effort put in defining the indicator toolbox and assessing its quality.

Many thanks for this supportive feedback.

About possible alternative datasets or other possible data analysis patterns: surely they are possible and reviewers have given relevant suggestions on this.

Nevertheless, authors have made a set or rational methodological choices both in selecting data and applying statistical tools and have been very transparent in explaining the reasons of their choices. Provided that these choices are convincing and scientifically sound (as I think they are), I believe they should be respected. Of course other pathways, including the application of methods borrowed from epidemiology could have been possible, but this could be the subject of a paper from another group or even from the authors themselves in future. Here the question is if the methodology and the input data chosen answer the research question posed, and in my opinion the authors mostly succeeded in this.

Many thanks. We have implemented further revisions to better clarify our methodological choices. In particular, we now clarify, in the main text, why Lasso was (in our view) the best method for the purposes of the analyses we intended to do; we also added some references demonstrating that this is a state-of-the art method widely used in similar situations in the current literature (see the section entitled ‘Multivariate lasso analysis, 1st paragraph, p. 8-9)

Even here, my main concern has been already voiced by Rev #4 noticing: “It seems that the authors are observing the relationships between CO2 and material stock indicators observed from the difference among countries, and then expecting that, for each country, change in the material stock pattern indicators will cause CO2 emissions to go down. I think that the authors cannot reach this conclusion.” Again, I agree with the rephrasing of this part of the paper made by the authors, that I consider acceptable for the publication.

Many thanks. In our revisions, we now respond to this concern by explicitly mentioning the need for (future) additional work based on temporal trajectories in the patterns of built structures; indeed, data that could support such studies are gradually becoming available.

On the contrary, I was disappointed by the very last sentence in the conclusions: “The area and patterns of built-up land emerge as important and novel entry point for efforts at reducing energy demand and CO2 emissions, which could contribute to limiting the reliance on contested CO2 reduction options such as bioenergy with carbon capture and sequestration”

Citing BECCS in the last sentence, without providing context and summarizing a complex debate in two lines and a single citation does not do justice to the authors' capacity of analyzing complex issues they have shown throughout the paper.

Indeed, it is true that BECCS can be a questionable source of energy, but it is also true that several other energy sources (I would say, EVERY energy source) has pros and cons, and I do not understand why open a view on such a complex debate in the last two lines of a good paper focused on totally different topic. On summary, I strongly suggest to remove this sentence or to give it a more general meaning.

Many thanks. We have strongly revised the last paragraph (p. 12) to better align conclusions with the results of our analysis, thereby omitting any reference to BECCS. In response to the comments by the editor, we also clarify policy implications of our work.